# MS-YOLOv8-Based Object Detection Method for Pavement Diseases

**DOI:** 10.3390/s24144569

**Published:** 2024-07-14

**Authors:** Zhibin Han, Yutong Cai, Anqi Liu, Yiran Zhao, Ciyun Lin

**Affiliations:** 1School of Transportation, Jilin University, Changchun 130022, China; hanzb1721@mails.jlu.edu.cn (Z.H.); caiyt1721@mails.jlu.edu.cn (Y.C.); liuaq1721@mails.jlu.edu.cn (A.L.); zhaoyr1721@mails.jlu.edu.cn (Y.Z.); 2Jilin Engineering Research Center for Intelligent Transportation System, Changchun 130022, China

**Keywords:** pavement diseases, object detection, deformable large kernel attention, multi-scale dilated attention

## Abstract

Detection of pavement diseases is crucial for road maintenance. Traditional methods are costly, time-consuming, and less accurate. This paper introduces an enhanced pavement disease recognition algorithm, MS-YOLOv8, which modifies the YOLOv8 model by incorporating three novel mechanisms to improve detection accuracy and adaptability to varied pavement conditions. The Deformable Large Kernel Attention (DLKA) mechanism adjusts convolution kernels dynamically, adapting to multi-scale targets. The Large Separable Kernel Attention (LSKA) enhances the SPPF feature extractor, boosting multi-scale feature extraction capabilities. Additionally, Multi-Scale Dilated Attention in the network’s neck performs Spatially Weighted Dilated Convolution (*SWDA*) across different dilatation rates, enhancing background distinction and detection precision. Experimental results show that MS-YOLOv8 increases background classification accuracy by 6%, overall precision by 1.9%, and mAP by 1.4%, with specific disease detection mAP up by 2.9%. Our model maintains comparable detection speeds. This method offers a significant reference for automatic road defect detection.

## 1. Introduction

As the economy continues to develop, the mileage of highways open to traffic in various countries is increasing, making highway travel an important mode of transportation in people’s lives. However, with the extensive and frequent use of highways, pavement diseases, such as cracks and potholes, frequently occur and have become a major factor affecting traffic conditions. Some roads experience a decline in service levels due to these pavement diseases. Disease detection is the first step in highway maintenance, making the exploration of road disease detection methods imperative. In road disease detection, traditional manual detection methods are inefficient, costly, and highly subjective to maintenance personnel [1], prompting scholars to explore the application of machine learning algorithms in road disease detection in recent years, aiming to achieve an efficient and intelligent pavement maintenance process. Despite many significant achievements, there are still numerous issues to be resolved in the field of automatic road disease detection. Firstly, the current algorithms are unstable in detecting multi-scale targets. Models need to extract features at different depths for targets of varying sizes, thereby requiring different processing strategies. Secondly, current algorithms face difficulties in detecting small-scale targets because small targets occupy fewer pixels in images, leading to the potential loss or neglect of detailed information about small targets. Thirdly, current algorithms face challenges in handling complex scenes, such as cluttered backgrounds and lighting, which can greatly interfere with target detection; for instance, branches in the background might be mistaken for cracks in road disease detection. At the same time, the target area of road disease often accounts for only a small part of the image, so a large amount of background information is introduced for the same disease category, different background information will make it impossible for us to obtain a unified disease characterization, and there may be similar background information between different categories of diseases, so that the model can not distinguish between the two [2]. To address these challenges, various foundational algorithms and models have been developed. For example, the widely used one-stage object detection algorithm “YOLO (You Only Look Once)” series in current research transforms object detection into a regression problem, predicting bounding boxes and class probabilities directly from image pixels. To improve multi-scale issues, the YOLO structure adopts an SPPF pyramid structure, incorporating multi-level feature pooling operations to capture spatial information at different scales. Additionally, to address the issue with small targets, a fourth small-target detection head has been introduced in YOLO. As a typical one-stage object detection algorithm, the YOLO series features fast detection speed and real-time performance, making it suitable for real-time scenarios in road disease detection. Currently, many studies have made improvements to the YOLO model to optimize for the aforementioned issues and enhance detection accuracy.

To better address the issues of background clutter and multi-scale challenges in road disease detection, this paper selects the highly accurate, frequently modified, and recently updated YOLOv8 as the base model, and integrates various model improvement algorithms for experimentation. Specifically, this study introduces an advanced model named MS-YOLOv8. The main objective of the research is to modify and optimize the YOLOv8 model to enhance its detection capabilities for multi-scale and complex background pavement diseases. The proposed modifications are as follows:

Firstly, the C2f layer in specific positions of the backbone network is replaced with C2f-DLKA, introducing the Deformable-LKA (Large Kernel Attention) mechanism into the initial C2f layer, enabling the model to more effectively extract features of complex multi-scale targets.

Additionally, inspired by similar innovations, the SPPF module in the backbone network is replaced with SPPF-LSKA. By incorporating the improved Large Kernel Attention module LSKA into the SPPF module, the model’s ability to aggregate features across multiple scales is enhanced.

Lastly, by adding Multi-Scale Dilated Attention (MSDA) to the network’s neck, the model’s adaptability to changes in the target disease scale is improved without increasing detection costs or introducing additional parameters.

The remaining sections of this paper are structured as follows: Section 2 provides a detailed introduction to the network structure of YOLOv8 and the improved DL-YOLOv8; Section 3 presents the experimental process, including burn-in tests and comparative experiments, the results of which verify the effectiveness of the improvements; and Section 4 summarizes the research findings and discusses potential future work.

## 2. Related Work

The use of machine learning methods for pavement disease detection has a long history. Before the emergence of novel deep learning algorithms, traditional machine learning methods were widely innovated and utilized. For instance, Cui [3] proposed an automatic pavement disease detection method using random forests, introducing features such as scale, color, and gradient into the model. Sulistyaningrum [4] employed the Support Vector Machine (SVM) algorithm to achieve linear classification detection of three types of pavement diseases: alligator cracking, potholes, and cracks. Zhang [5] combined SVM and boosting methods to classify pavement diseases into cracks and non-cracks, a binary classification approach that represents an initial integration of deep learning algorithms with pavement disease detection.

Currently, as a subfield of machine learning, deep learning algorithms are extensively used in pavement disease detection through three approaches: image classification, semantic segmentation, and object detection. Compared to traditional machine learning algorithms, deep learning can automatically extract deep, advanced image features through multiple layers of convolution and pooling without human intervention. Additionally, the end-to-end training characteristic of deep learning algorithms allows for global optimization through methods like gradient descent, rather than stage-by-stage optimization adjustments. Furthermore, deep learning algorithms can process and train large-scale data without the efficiency concerns associated with algorithms like SVM.

Simultaneously, more scholars are exploring improvements based on initial model frameworks. In semantic segmentation, the FCN (Fully Convolutional Network) method is the most popular. U-Net, an image segmentation model based on FCN with superior performance, is also widely used. Jenkins [6] proposed a pavement disease segmentation model primarily structured around U-Net. Inspired by U-Net, Majidifard [7] combined YOLO with U-Net to determine both the location and the extent of pavement damage. Lau [8] used a transfer learning approach, incorporating the pre-trained ResNet-34 neural network into the encoder, enhancing the model’s performance to 96%. In object detection, R-CNN, SSD, and YOLO are noted for their convenient and rapid detection capabilities. For example, Chang [9] used a CNN model to predict cracks; Asadi [10] prepared a pavement crack dataset using a global shutter RGB-D sensor mounted on a vehicle and experimented with various backbones in the Faster R-CNN and SSD detection frameworks, achieving high-accuracy crack detection. Yu [11] improved the YOLOv4 to achieve superior pavement disease detection accuracy, with a speed comparable to the Faster R-CNN algorithm.

However, significant scale differences in pavement diseases, high background interference, and varied disease shapes continue to challenge scholars. These issues can cause the model to learn disease features inadequately or mistakenly recognize cluttered backgrounds as pavement diseases, preventing ideal detection outcomes. To mitigate these impacts on model accuracy, past research has tackled improvements from various angles by introducing algorithms at different points within the model framework. For instance, Chen [12] replaced the backbone of YOLOv4 with MobileNetV2, enhancing the model’s ability to recognize complex shapes. Qu [13] addressed multi-scale target issues by incorporating deep feature maps into shallow ones, fully utilizing advanced deep semantic features. Liu [14] changed the detection head of YOLOv3 to four, improving the model’s accuracy in detecting small targets. Liu [15] proposed integrating Dynamic Snake Convolution with a Context Attention Mechanism (CAAM) into YOLOv8, enhancing the model’s ability to recognize targets in complex backgrounds, especially when targets are obstructed. Peng [16] proposed a global Receptive Field Pyramid (SPPF) algorithm, enhancing the YOLOX model’s detection accuracy on multi-scale data by maintaining rich inter-channel information.

In summary, using machine learning and deep learning algorithms for automatic pavement disease detection has become a current trend. By introducing different base models and making improvements, many researchers have contributed to the field of pavement disease detection, enhancing the model’s accuracy in detecting pavement diseases. However, current research largely focuses on integrating various algorithms into base models to address specific issues in the detection process, such as the small target problem, and lacks comprehensive optimization of models to simultaneously address issues like significant scale differences, high background interference, and varied disease shapes. Therefore, to more accurately enhance the model’s detection capabilities and improve its performance in real-world situations, this paper proposes the MS-YOLOv8 model based on the YOLOv8 model, achieving comprehensive improvement and optimization for three detection issues, further enriching the research in the field of pavement disease detection and laying the groundwork for future exploration.

## 3. Method

### 3.1. YOLOv8 Algorithm Overview

YOLOv8 is an efficient real-time object detection model that consists of inputs, a backbone network, neck architecture, and detection heads. It is a deep convolutional neural network with 53 convolutional layers [17]. In the detection process, the input images to be analyzed first pass through the backbone network, where features are initially extracted using convolutional layers (Conv), shortcut connections (C2f), and an SPPF structure. Subsequently, the feature maps enter the neck architecture, where a series of convolutional operations and upsampling processes are performed. This stage enables the fusion of shallow and deep features, facilitating more effective detection of targets. The fused multi-scale feature maps are then fed into three different detection heads, each specialized for handling targets of varying sizes, such as, for example, the detection head for large targets focuses on deeper feature information. Ultimately, each detection head outputs a series of prediction boxes, which contain the detected targets. This paper builds on the YOLOv8 model, utilizing an improved version of YOLOv8 to identify and detect four types of pavement diseases. The specific model training and detection processes are the same, as illustrated in Figure 1.

### 3.2. Improved YOLOv8 Network Structure

In this paper, the MS-YOLOv8 framework is proposed based on the YOLOv8 model. In view of the characteristics of small targets and multi-scale pavement diseases, we have optimized the backbone and neck of the original YOLOv8 to enhance its performance in detecting pavement diseases. The improvement strategy involves integrating three special mechanisms into the YOLOv8 framework:Deformable LKA: This mechanism modifies certain C2f layers in the backbone and neck using a Deformable Large Kernel Attention mechanism. Deformable LKA combines large convolutional kernels with deformable convolutions, employing depth-wise and depth-wise dilated convolutions to construct large kernels. This allows the network to learn features within a larger receptive field. The deformable convolution adjusts the sampling positions of standard convolutions by adding additional offsets, enhancing the model’s capability to capture irregular shapes and sizes. The combination of these features improves the network’s ability to detect small and irregularly shaped targets. We redesign the C2f at different positions in the original model, introduce the DLKA mechanism combining variability convolution and Large Kernel Attention mechanism into the C2f to obtain a new C2f_DLKA structure, and verify through experiments that the modification of the C2f structure at the selected position in the paper can make the model optimal.SPPF-LSKA: If the Large Kernel Attention mechanism continues to be used to expand the receptive field, the network can obtain more global context information and reduce the influence of complex background on disease identification. However, because the feature map becomes smaller and contains more semantic features and relatively few detailed features of the original image after three consecutive maximum pooling operations in the SPPF structure, the DLKA mechanism with deformable convolution is no longer suitable for use. Consider that the complexity of the model should not be excessively increased, replacing the original SPPF structure in YOLOv8 with SPPF-LSKA; this mechanism decomposes the deep convolutional layer’s 2D kernels into cascaded horizontal and vertical 1D kernels. Compared to the traditional large kernel and attention mechanism (LKA), this strategy efficiently reduces computational complexity and memory demands while maintaining effective image processing capabilities. When combined with the original SPPF structure, it enhances the SPPF module’s ability to aggregate features across multiple scales.Multi-Scale Dilated Attention: This feature was introduced into the neck of YOLOv8 for further processing of the three feature maps of the output. The feature maps’ channels are divided into several different heads, each undergoing a Sliding Window Dilated Attention (*SWDA*) operation with varying dilation rates set for different heads. This helps the network to aggregate semantic information across various scales. All outputs are then concatenated and passed through a linear layer for feature aggregation. This design enables the model to understand images across different scales, capturing both local details and broader contextual information, thereby enhancing the model’s performance. The network structure of MS-YOLOv8 is illustrated in the following Figure 2.

Our research provides a lightweight model with high accuracy, which is still superior on different datasets, and the lightweight of the model allows it to be installed on mobile devices and vehicle-mounted tools, which provides a feasible scheme for the task of pavement disease detection and is of great significance for the real-time detection and timely maintenance of pavement defects.

#### 3.2.1. DLKA Mechanism

DLKA stands for Deformable Large Kernel Attention, which combines the attention mechanism with large convolutional kernels and deformable convolutions. Large kernels are mechanisms capable of capturing extensive contextual information within images, offering a receptive field similar to that of self-attention mechanisms. These are constructed using fewer parameters and reduced computational complexity through the use of depth-wise convolutions, depth-wise dilated convolutions, and 1 × 1 convolutions. Deformable convolutions introduce an offset to the sampling positions in standard convolution operations, allowing the kernel to expand considerably during training. This flexible and adaptable kernel can better conform to the shapes and sizes of objects during sampling, thereby enhancing the ability to capture objects of irregular shapes and sizes. The structure of this mechanism is illustrated in Figure 3 [18]. From the diagram, it can be seen that the module consists of multiple convolutional layers, including standard convolutions, depth-wise convolutions with offsets, and depth-wise dilated convolutions. For an input image with dimensions *H × W × C*, the equations to construct *k* × *k* size depth-wise and depth-wise dilated convolutional kernels are as follows:(1)$$DW=\left(2d-1\right)\times \left(2d-1\right)$$
(2)$$DW-D=\frac{k}{d}\times \frac{k}{d}$$

In this case, *k* represents the size of the convolutional kernel, and *d* represents the dilation rate.

Standard convolution operates by performing convolution calculations with a fixed size and shape of the convolutional kernel over the input feature map. The kernel then slides across the input in a defined manner to compute the entire output feature map. Thus, for any point *P*_0_ on the input feature map, the convolution operation can be represented as follows:(3)$$y\left(P_{0}\right)=\underset{P_{n}\in R}{\sum }w\left(P_{n}\right)\times x\left(P_{0}+P_{n}\right)$$
where $P_{n}$ represents the offset of each point in the convolutional kernel relative to the center point, $w\left(P_{n}\right)$ indicates the weight of the convolutional kernel at the corresponding position, $x\left(P_{0}+P_{n}\right)$ denotes the element value at the position $\left(P_{0}+P_{n}\right)$ on the input feature map, and $y\left(P_{0}\right)$ represents the element value at the position $P_{0}$ on the output feature map, obtained through the convolution of the kernel with the input feature map.

After introducing deformable convolutions, an additional convolution is used to learn offsets from the feature map, resulting in offsets. Based on these offsets, the sampling points of the convolutional kernel on the feature map are shifted. Deformable convolution introduces an offset into the aforementioned formula as follows:(4)$$y\left(P_{0}\right)=\underset{P_{n}\in R}{\sum }w\left(P_{n}\right)\times x\left(P_{0}+P_{n}+\Delta P_{n}\right)$$

In this, $\Delta P_{n}$ represent the offsets. 

The size and dilation rate of the convolutional kernel responsible for calculating the offsets are consistent with those of the corresponding convolutional layer. When the position adjusted by the offsets does not correspond to actual pixel points on the feature map, bilinear interpolation is used to obtain the pixel values after the offset. Bilinear interpolation can be expressed with the following formula:(5)$$x\left(P\right)=\underset{q}{\sum }G\left(q,p\right)\times x\left(q\right)$$

The DLKA is used to optimize the C2f structure in YOLOv8 by replacing the second Conv convolution in the bottleneck section of the original c2f with DLKA. The new c2f-DLKA structure compared to the original c2f structure is shown in Figure 4 below. This improvement enhances the model’s ability to capture multi-scale targets.

#### 3.2.2. MSDA Mechanism

The scale and size of pavement diseases vary greatly, and the size of the receptive field significantly impacts the detection results: if the size of an object exceeds the boundaries of the receptive field, it will lead to insufficient feature extraction; conversely, if the receptive field is too large compared to the actual size of the object, the background information can negatively affect recognition. Therefore, using a fixed-size convolutional kernel for the detection of pavement diseases is not effective. MSDA divides the feature map’s channels into n different heads and performs multi-scale dilated convolutions at different dilation rates within these heads, referred to as *SWDA*. This mechanism enables the model to capture image features at various scales. The following Figure 5 shows the receptive when the dilation rate r = 3, and Figure 6 demonstrates the specific working principle MSDA. Replace “dilation” for dilation rate, and “pavement diseases” for diseases in the translation.

As can be seen, this mechanism first divides the feature map’s channels into several groups, with each group assigned to a corresponding attention head. For the different attention heads, various dilation rates (such as 1, 3, 5) are used, allowing the receptive fields to cover areas of different sizes. The use of different dilation rates in the groups can make the model focus more on different parts of the input data, enhancing the extraction of multi-scale features while also broadening the information gathered. Subsequently, self-attention operations are performed in the different heads, employing a method of weighting different image features to make the model pay more attention to beneficial information. The MSDA mechanism effectively addresses the issue with traditional attention mechanisms that focus on local attention and overlook important information at a distance.

The *SWDA* operation is repeated for each attention head using the following formula:(6)$$h_{i}=SWDA\left(Q_{i},K_{i},V_{i},r_{i}\right),1\leq i\leq n$$

In the formula, $Q_{i},K_{i},V_{i}\;$ represent the query (*Q*), key (*K*), and value (*V*) for each head, respectively. $r_{i}$ indicates the dilation rate corresponding to each head (for example, $r_{1}\;$ = 1,$\;r_{2}\;$ = 3), and $h_{i}$ is the output for each head.

The specific operation of *SWDA* can be expressed as follows:

For the position (i, j) in the original feature map before splitting, the output component $X_{ij}$ of the *SWDA* operation is defined as follows:
(7)$$X_{ij}=Attention\left(Q_{ij},K_{r},V_{r}\right)=Softmax\left(\frac{Q_{ij}K_{r}^{T}}{\sqrt{d_{K}}}\right)\times V_{r},\;1\leq i\leq W,1\leq j\leq H$$


In the formula, H and W represent the height and width of the feature map, $Q_{ij}$ represents the query vector at position (i,j), and $K_{r},\;V_{r}$ represent the key and value vectors selected from K,V,  respectively. The SoftMax function is used for normalization. For a specific position (i,j), a set of keys and values at specific coordinates ($i^{\prime },j^{\prime }$) is selected to perform the self-attention calculation as follows:
(8)$$\left\{\left(i^{\prime },j^{\prime }\right)|i^{\prime }=i+p\times r,j^{\prime }=j+q\times r\right\}\;-\frac{w}{2}\leq p,q\leq \frac{w}{2}$$


In the formula, w represents the size of the sliding window.

After obtaining multi-scale features through the self-attention mechanism, MSDA effectively integrates these features via a linear layer (linear). This allows the model to not only perceive local details of an image but also understand its overall structure, thereby enabling a more comprehensive understanding of the image content [19].

#### 3.2.3. SPPF-LSKA Mechanism

The traditional Large Kernel Attention (LKA) module has demonstrated excellent performance in Visual Attention Networks (VAN), but it faces challenges related to high computational and memory demands when dealing with large-sized convolutional kernels. The LSKA mechanism addresses these issues by decomposing the 2D convolutional kernels of deep convolution layers into two cascaded 1D convolutional kernels. This decomposition significantly reduces computational complexity and the number of parameters. Additionally, performing convolution operations in two steps with the cascaded 1D convolutional kernels can achieve similar effects to the original large-sized 2D convolutional kernels [20]. Incorporating the LSKA mechanism into the SPPF structure, the modified SPPF-LSKA and the specific structure of LSKA are illustrated in the following Figure 7.

In the LSKA structure depicted in the diagram, the two consecutive DW-Conv and DW-D-Conv represent cascaded 1D convolutional kernels obtained by splitting the 2D depth-wise convolution and depth-wise dilated convolution kernels. To achieve effects similar to those of a k×k convolutional kernel, the sizes of the four 1D convolutional kernels from left to right should be as follows:
(9)$$1\times \left(2d-1\right),\left(2d-1\right)\times 1\;1\times [\frac{k}{d}],[\frac{k}{d}]\times 1$$
where d represents the dilation rate. When an input feature map $F\in R^{C\times H\times W}$ with channel number C and dimensions H and W is processed, the output of LSKA can be obtained through the following formula:(10)$${\bar{Z}}^{C}=\underset{H,W}{\sum }W_{\left(2d-1\right)\times 1}^{C}*(\underset{H,W}{\sum }W_{1\times \left(2d-1\right)}^{C}*F^{C})$$
(11)$$Z^{C}=\underset{H,W}{\sum }W_{\left(\frac{k}{d}\right)\times 1}^{C}*(\underset{H,W}{\sum }W_{1\times \left(\frac{k}{d}\right)}^{C}*{\bar{Z}}^{C})$$
(12)$$A^{C}=W_{1\times 1}*Z^{C}$$
(13)$$F^{C}=A^{C}⭙F^{C}$$

${\bar{Z}}^{C}$ represents the output of the feature map after passing through the first two cascaded 1D convolutional kernels, which capture local spatial information. $Z^{C}$ is the output after passing through the next two 1D convolutional kernels derived from depth-wise dilated convolutions, responsible for capturing global spatial information. Subsequently, a 1 × 1 convolutional kernel is used to perform a convolution operation on this output to obtain AC. This AC is then combined with the input feature map FC through a residual connection operation (Hadamard product) to produce the output of the LSKA.

In MS-YOLOv8, we have incorporated the LSKA mechanism into the SPPF structure. This introduction slightly increases the computational load and model parameters but effectively mitigates the limitations associated with using small convolutional kernels and enhances the ability of the SPPF module to aggregate features across multiple scales.

## 4. Experiment

### 4.1. Dataset Preparation

As an object detection algorithm, yolov8 has some requirements for the resolution, angle, and lighting conditions of the input image.

YOLOv8 can process images of different resolutions, and the common input sizes are 416 × 416, 512 × 512, 640 × 640, etc.;Images from different camera angles should be included in the training data to enhance the robustness of the model, such as side, front, top, and up;Lighting conditions have a significant impact on the detection performance of YOLOv8. A good training dataset should include images acquired under different lighting conditions to ensure that the model performs well in different lighting environments.

It is recommended to use high-quality camera equipment to avoid blurry images.

The road damage dataset RDD2022, released by the Crowdsensing-based Road Damage Detection Challenge (CRDDC 2022), comprises four types of pavement diseases, as listed in Table 1 below. It includes over 55,000 instances of road damage from six countries—Japan, India, Czech Republic, Norway, USA, and China. The RDD2022 dataset incorporates rich road environments from different countries, enhancing the universality and robustness of our model. The data collection methods include smartphones mounted on vehicles and high-resolution cameras inside windshields [21], making our model well suited for low-cost automatic road damage detection.

Before formal training, all images undergo certain preprocessing steps. Firstly, we use Python to remove road images that lack annotations. Secondly, we use Labelimg to review annotated images and filter out those with blurred damage or unclear annotations. Since the detection targets include cracks in “transverse” and “longitudinal” orientations, commonly used methods like auto-orient are avoided as they may change the direction of cracks, leading to type confusion. For the entire dataset, we select some images for Mosaic augmentation, which improves the model’s generalization ability and accuracy in background recognition by combining fragments of multiple images into new image data. After processing, the dataset contains a total of 43,992 images, with a relatively even distribution across the participating countries as Figure 8, and the distribution of various types of diseases is relatively uniform as Figure 9. Finally, the dataset is divided into training, validation, and testing sets in a 7:2:1 ratio using a random sampling method.

### 4.2. Experimental Environment Configuration

Our experiments were conducted on the AutoDL cloud platform, and the specific environment configuration information is shown in Table 2.

In terms of hardware, we utilized advanced components for high performance, such as an AMD EPYC 9754 CPU and an RTX 3090 GPU. Regarding software, we chose Ubuntu 20.04 as the operating system, Python 3.8 as the programming language, PyTorch 1.11.0 as the deep learning library, and CUDA 11.3 and cuDNN for accelerating and optimizing the computation, training, and inference processes of deep learning.

### 4.3. Evaluation Metrics

In order to evaluate the results of the target detection model training, the following are important assessment metrics. Based on these metrics, we can better understand the performance exhibited by the model in different aspects, and use these metrics to choose the most suitable model. For example, if a region has a variety of pavement disease types, the mAP (mean Average Precision) metric becomes more important; if the roads in the area are dangerous, where any pavement disease could lead to severe consequences, the recall metric becomes more important. Specific metrics are described in Table 3 below.

Where TP (True Positive) represents the number of instances correctly identified as positive by the model. FN (False Negative) represents the number of positive instances that the model incorrectly identified as negative. FP (False Positive) represents the number of negative instances that the model incorrectly identified as positive. N is the total number of classes and $AP_{i}$ represents the average precision for total classes.

### 4.4. Ablation Study

The purpose of conducting an ablation study is to verify the positive effects of improvement strategies on model training outcomes and to test for synergistic interactions between different modules. This study includes three modules: DLKA, MSDA, and SPPF-LSKA. The results are presented in Table 4 below, which shows the outcomes of experiments using different modules.

Experimental results indicate that when the DLKA module is incorporated into the network module separately, the accuracy slightly decreases, while the recall rate increases by 0.41%, mAP@0.5 increases by 0.42%, and mAP@0.5:0.95 increases by 0.1%, showing a small improvement. After individually introducing the MSDA module, the accuracy improves by 1.57%, the recall rate by 0.41%, mAP@0.5 by 0.5%, and mAP@0.5:0.95 by 0.24%. When both MSDA and DLKA modules are introduced simultaneously, the accuracy increases by 1.68%, recall by 0.18%, mAP@0.5 by 0.72%, and mAP@0.5:0.95 by 0.27%.

When all three improvement measures are finally introduced, the model’s accuracy increases by 1.83% and mAP@0.5 by 1.37%, indicating that the inclusion of DLKA, MSDA, and SPPF-LSKA modules enables the model to better adapt to the detection of multi-scale objects, and the modules do not inhibit each other’s performance. Figure 9 displays the detection results of different types of diseases by the newly developed MS-YOLOv8 model.

Furthermore, to demonstrate the superiority of the improved YOLOv8 model in the task of road surface defect detection, we also present the detection results of other object detection algorithms (YOLOv3-tiny, YOLOv5n, YOLOv7-tiny, YOLOv8n, SSD) on the RDD2022 dataset. It can be seen that MS-YOLOv8 has the highest accuracy, significantly enhancing the capability of object detection (Table 5).

Then, the figures below show the specific results in many models. Image A (left) was captured by a vehicle-mounted device and is a typical example of complex background image data, which also features multi-scale characteristics of pavement diseases. Currently, various advanced detection models have presented the results shown on the left side of the image below. YOLOv5n and YOLOv7-tiny both exhibit different degrees of missed detections, with YOLOv7-tiny showing the most severe cases of false negatives. YOLOv8n, affected by the complex background and multi-scale targets, experiences significant false positive issues. However, the MS-YOLOv8 model proposed in this study demonstrates excellent performance in this scene (Figure 10), accurately identifying the number and types of pavement diseases with minimal interference from the complex background.

Image B (right) was taken with a smartphone, which provides high clarity and good lighting and angle, and there is a small target pavement disease in the bottom left corner of the image, which is useful for testing the model’s ability to recognize small targets. Similar to the results for Image A, the YOLOv8n model also exhibits false positives, showing multiple bounding boxes for the same disease (Figure 11). YOLOv7-tiny misses the small pavement disease in the bottom left corner (Figure 12), and YOLOv5n exhibits lower detection performance, missing the obvious cracks in the image (Figure 13). In contrast, the MS-YOLOv8 model proposed in this study accurately identifies the locations of three pavement diseases and produces bounding boxes with high confidence, demonstrating that the MS-YOLOv8 model also significantly enhances the recognition capability for small targets.

These specific detection image examples adequately demonstrate the good detection capability and accuracy of the MS-YOLOv8 model proposed in the research, with strong resistance to interference from complex backgrounds, and superior recognition ability for multi-scale and small target pavement diseases compared to previous models.

From the experimental results (Table 6), it can be seen that the indicators of MS-YOLOv8 are lower than those on the RDD2022 dataset, which may be due to the small number of images in the newly selected dataset, but compared with the initial model of YOLOv8, the performance of MS-YOLOv8 is still better than it, which shows that our model still has superiority on different datasets.

## 5. Results

From the experimental results, it can be observed that our targeted improvements on the YOLOv8 model address the issues of poor performance in detecting small objects, multi-scale objects, and objects that closely resemble the background in object detection algorithms. The optimized MS-YOLOv8 model demonstrates superiority in road damage detection tasks. By incorporating the Deformable Large Kernel Attention mechanism to modify the c2f architecture of YOLOv8, Deformable Large convolutional kernels are introduced, allowing the shape of the kernels to flexibly adapt to the actual shapes of objects, thereby aiding in the detection of irregular road damages.

Following a similar approach, the SPPF structure is replaced with the SPPF-LSKA structure, constructing a 2D deep convolution layer using a series of 1D convolution kernels in sequence. This design enhances the model’s feature extraction capability without significantly increasing the model’s parameters and computational complexity. Finally, the MSDA mechanism is introduced into the neck of YOLOv8, employing convolutional kernels with varying dilation rates to perform sliding window attention mechanisms in different heads of the feature map, and then integrating the resulting features. Such a design allows the model to understand images at different scales without significantly increasing the model’s complexity.

The final MS-YOLOv8 model maintains the highest detection accuracy while having a smaller model volume, making it less demanding on the operating device and enabling deployment on mobile or embedded devices, which is beneficial for its application in real-time road damage detection. In subsequent work, research will focus on how to make the model even more lightweight while maintaining detection speed to better suit real-time road damage detection tasks. Additionally, the figure above (Figure 14) displays the detection results of the newly proposed MS-YOLOv8 model on different road image data, demonstrating that MS-YOLOv8 accurately identifies road damages with high confidence.

## 6. Conclusions

After redesigning c2f and SPPF using the Large Kernel Attention mechanism, we expanded the receptive field of the model so that the network could obtain more global context information and reduce the influence of complex background on disease detection results.

This project, based on the use of deep learning methods for pavement disease detection, addresses issues of multi-scale diseases, small target detection, and difficulty distinguishing diseases from the background by introducing the MS-YOLOv8 model. This model builds on the YOLOv8 architecture, modifying its backbone and neck components, and integrating three algorithms and mechanisms (DLKA, MSDA, LSDA) into one cohesive system. Experiments have demonstrated that the MS-YOLOv8 model achieves good detection accuracy for diseases. Compared to the YOLOv8 model, the probability of correctly classifying disease images against their backgrounds has increased by 6%, overall precision (P) has improved by 1.9%, and overall mAP has increased by 1.4%. For specific types of road diseases, the mAP has increased by as much as 2.9%. Additionally, compared to models such as YOLOv3-tiny, YOLOv5n, and YOLOv7-tiny, there is a significant improvement in detection accuracy.

This project further enriches the field of road disease target detection based on deep learning by incorporating algorithmic improvements from other fields into road disease detection, providing a direction for improvements to existing models. Moreover, the enhanced accuracy of the model aids in advancing the implementation of automatic road disease detection, greatly ensuring road safety.

However, the current deep learning models based on YOLOv8 improvements still cannot completely accurately identify different types of road diseases, and the disease recognition process is still affected by background noise in the images being analyzed. Due to limitations in balancing and retaining the deep and shallow features extracted by deep learning models, issues of multi-scale and small targets have not been completely resolved. In the future, this model will be further improved, gradually attempting to use more advanced models (such as YOLOv9) as the base for training. While enhancing detection accuracy, the algorithm will also be optimized to make the model more lightweight—for example, by reducing model size, decreasing the number of floating points, and improving computational efficiency to achieve end-to-end efficient and accurate detection.

## Figures and Tables

**Figure 1 sensors-24-04569-f001:**
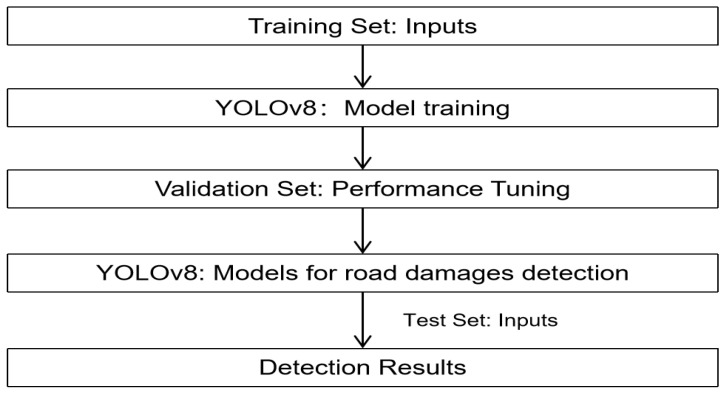
The flowchart of YOLOv8.

**Figure 2 sensors-24-04569-f002:**
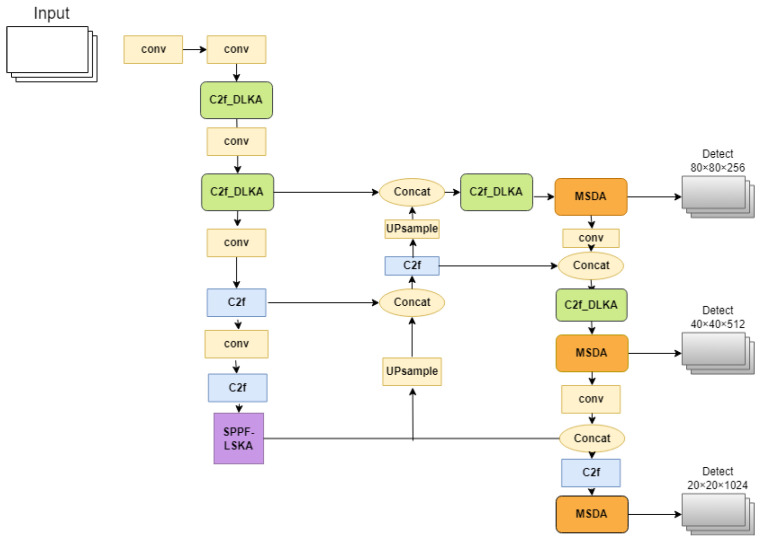
The overall network structure of MS-YOLOv8.

**Figure 3 sensors-24-04569-f003:**
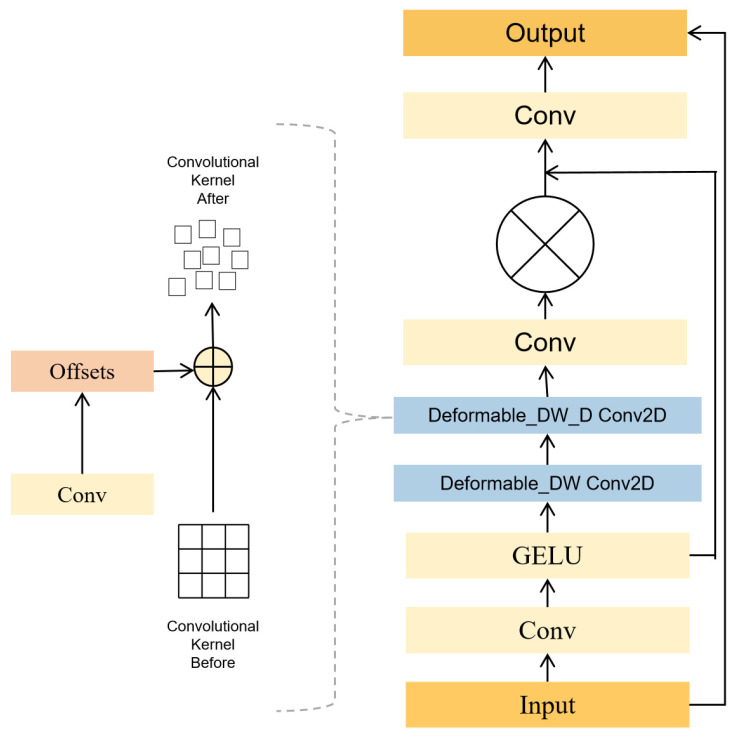
Comparative schematic diagram of C2f structure and C2f-DLKA structure.

**Figure 4 sensors-24-04569-f004:**
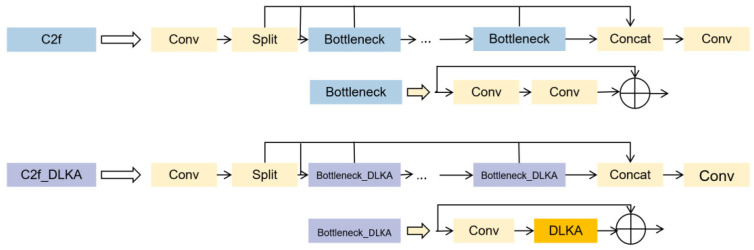
The schematic diagram of DLKA mechanism structure.

**Figure 5 sensors-24-04569-f005:**
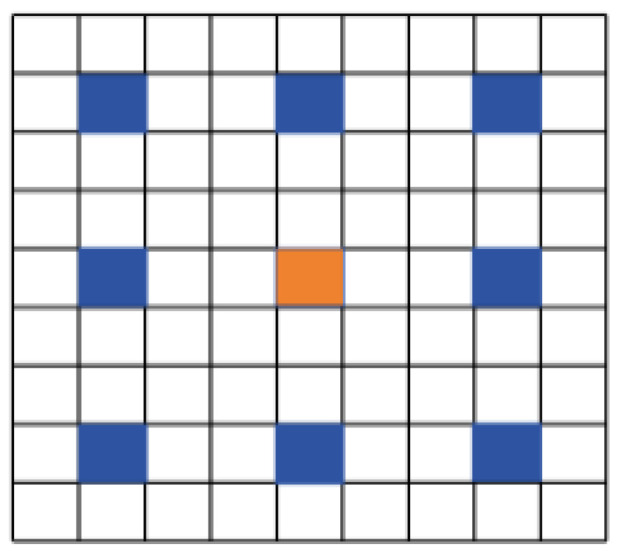
The schematic diagram of the receptive field at dilation rate r = 3. (This diagram shows how the empty attention of the sliding window works, where the orange box represents the center patch of the current attention, and the blue box represents the adjacent patch that is selected for attention calculation).

**Figure 6 sensors-24-04569-f006:**
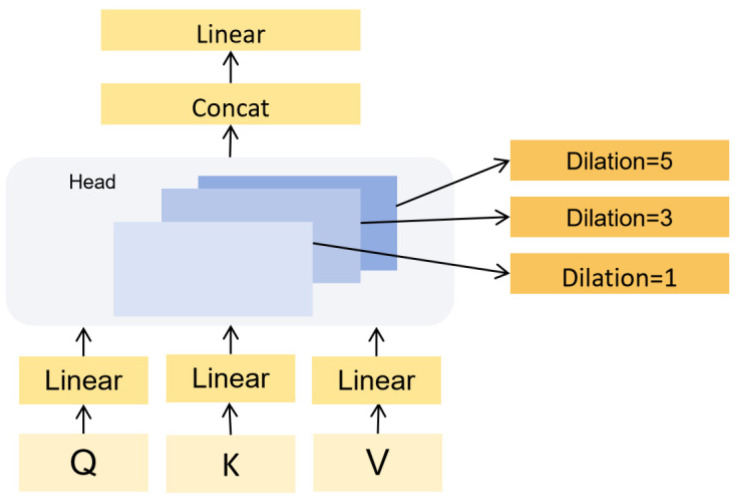
The schematic diagram of the working principle of MSDA.

**Figure 7 sensors-24-04569-f007:**
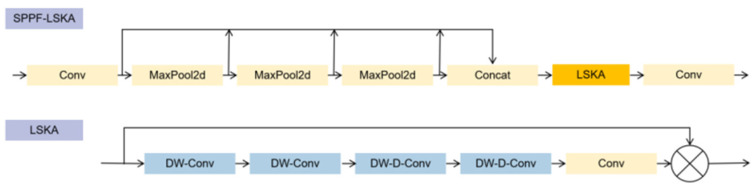
Comparative schematic diagram of SPPF-LSKA and SPPF structures. (The horizontal arrows indicate the order of the data flow, passing through the modules from left to right. The vertical arrows indicate the flow direction of the feature map after the pooling operation, and different pooling layers will produce feature maps of different scales, which will be stitched together).

**Figure 8 sensors-24-04569-f008:**
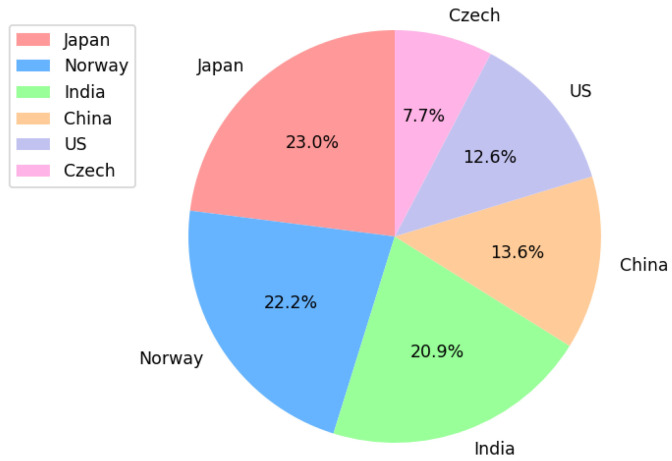
Proportional distribution of pavement disease images by country.

**Figure 9 sensors-24-04569-f009:**
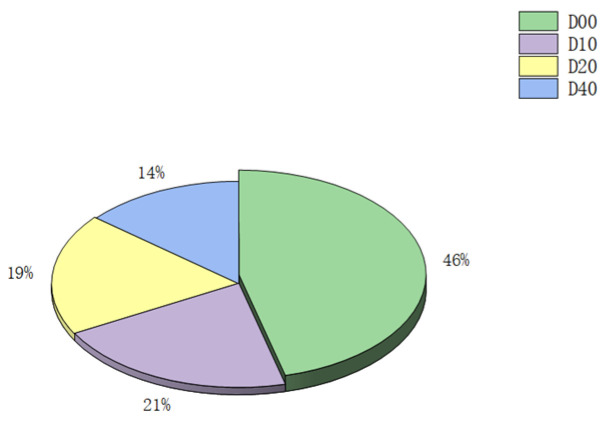
Proportions of types of diseases in the dataset.

**Figure 10 sensors-24-04569-f010:**
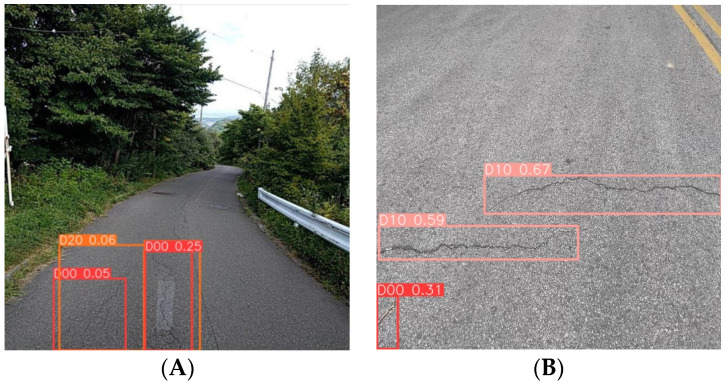
MS-YOLOv8 detection results. (**A**). On-board device capture, (**B**). Smartphone capture.

**Figure 11 sensors-24-04569-f011:**
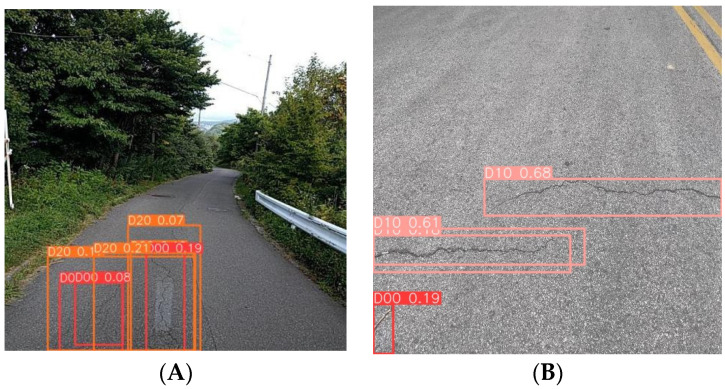
YOLOv8 detection results. (**A**). On-board device capture, (**B**). Smartphone capture.

**Figure 12 sensors-24-04569-f012:**
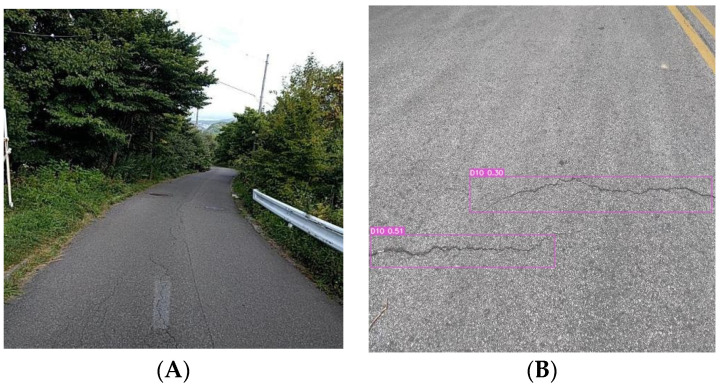
YOLOv7 detection results. (**A**). On-board device capture, (**B**). Smartphone capture.

**Figure 13 sensors-24-04569-f013:**
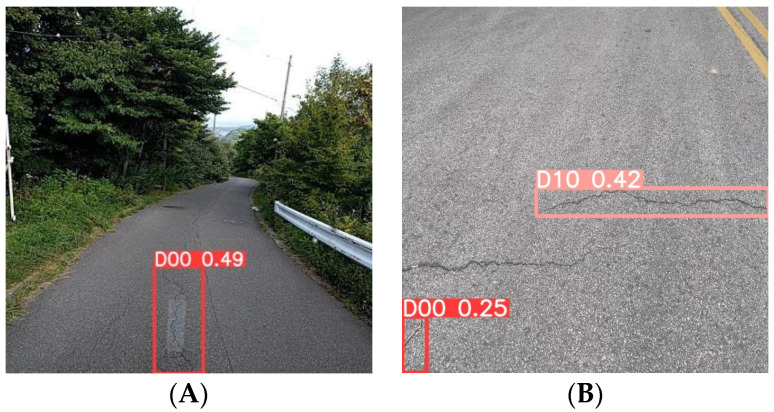
YOLOv5 detection results. (**A**). On-board device capture, (**B**). Smartphone capture.

**Figure 14 sensors-24-04569-f014:**
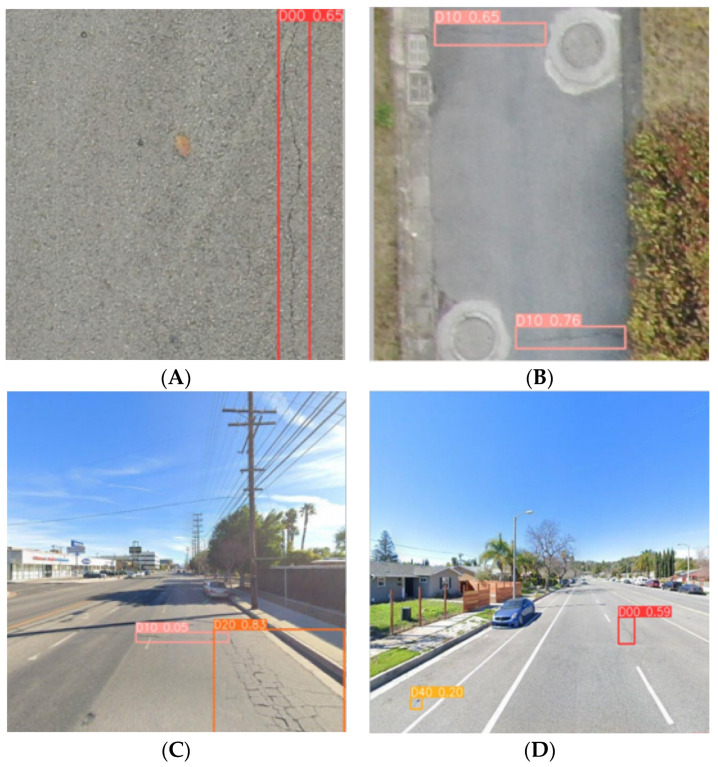
The detection results of the MS-YOLOv8 model for different types of diseases. (**A**). Longitudinal Crack, (**B**). Transverse Crack, (**C**). Transverse Crack and Alligator Crack, (**D**). Longitudinal Crack and Potholes.

**Table 1 sensors-24-04569-t001:** Types of pavement diseases in the RDD2022 dataset [22].

Class Name	Diseases
D00	Longitudinal Cracks
D10	Transverse Cracks
D20	Alligator Cracks
D40	Potholes

**Table 2 sensors-24-04569-t002:** Experimental environment configuration.

Configuration	Name	Specific Information
Hardware Environment	CPU	AMD EPYV 9754 128-Core Processor
	GPU	PTX 3090
	VRAM	24 GB
	Memory	60 GB
Software Environment	Operating System	Ubuntu20.04
	Python Version	3.8
	PyTorch Version	1.11.0
	CUDA Version	11.3
	cuDNN Version	7.6.5

**Table 3 sensors-24-04569-t003:** Description of evaluation metrics for the YOLOv8-based model.

Name	Calculation Method
R (Recall)	$\frac{TP}{TP+FN}$
Box P (Precision)	$\frac{TP}{TP+FP}$
F1	$\frac{P\times R}{P+R}\times 2$
mAP (mean Average Precision)	$\frac{1}{N}\overset{N}{\underset{i=1}{\sum }}AP_{i}$

**Table 4 sensors-24-04569-t004:** Results of the ablation study.

Model	DLKA	MSDA	SPPF-LSKA	P	R	Map@0.5	mAP@0.5:0.95
YOLOv8	×	×	×	0.630	0.527	0.571	0.295
	√	×	×	0.629	0.531	0.575	0.296
	×	√	×	0.646	0.531	0.576	0.297
	×	×	√	0.641	0.526	0.577	0.3
	√	√	×	0.647	0.529	0.578	0.297
	√	√	√	0.649	0.537	0.585	0.300

**Table 5 sensors-24-04569-t005:** Performance comparison of different models.

Model	mAP@0.5	mAP@0.5:0.95	F1
YOLOv3-tiny	0.450	0.176	0.499
YOLOv5n	0.534	0.242	0.564
YOLOv7-tiny	0.563	0.261	0.569
YOLOv8n	0.571	0.295	0.574
MS-YOLOv8	0.585	0.300	0.588
YOLOv8-SnakeVision	0.575	0.297	0.582

**Table 6 sensors-24-04569-t006:** Performance on other datasets.

Model	Precision	Recall	F1 Score	mAP@0.5	mAP@0.5:0.95
YOLOv8	0.444	0.357	0.396	0.354	0.160
MS-YOLOv8	0.449	0.403	0.425	0.371	0.174

## Data Availability

We used two publicly available datasets in our study, one from https://github.com/sekilab/RoadDamageDetector/?tab=readme-ov-file (accessed on 10 July 2024) and one from https://github.com/keaidesansan/Roadcrack_Dataset_2517.git (accessed on 10 July 2024).

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
