# Peer review of "MS-YOLOv8-Based Object Detection Method for Pavement Diseases"

_sensors, 2024, doi:10.3390/s24144569_

Round 1

Reviewer 1 Report

Comments and Suggestions for Authors

The paper introduces MS-YOLOv8, enhancing pavement disease recognition with mechanisms like DLKA, LSKA, and SWDA, improving detection accuracy and adaptability.Solving the following problems may help to improve the quality of this paper:

(1)Page 11, L305, [?], the reference is missing

(2)The challenges related to handling complex scenes, such as cluttered backgrounds and lighting, which can interfere with target detection, are not extensively discussed in the paper. 

(3)Although the MS-YOLOv8 model demonstrates improvements in background classification accuracy, precision, and mean Average Precision (mAP), the paper does not offer detailed insights into the limitations or potential drawbacks of the proposed algorithm, understanding the limitations of the model is essential for evaluating its suitability in various scenarios.

(4)There is a limited discussion on the generalizability of the proposed method across different pavement disease datasets or road conditions, only cracks were involved in the current dataset, further analysis on the model's performance variability with diverse datasets would enhance the paper's conclusions.

(5) The requirements on the photos, for example, the camera angle and resolution, are not discussed in the paper.

Comments on the Quality of English Language

The English is ok.

Reviewer 2 Report

Comments and Suggestions for Authors The authors consider the problem of identifying road surface defects and analyzing various neural network models to solve this task. The presented paper could potentially interest researchers in the recognition and development of road surface monitoring and control systems for transport systems.   However, a number of comments for the paper should be noted:   1. All available references are appropriate, but they are not enough. The authors do not refer to articles about the methods on which their methodology is based. In particular, why do the authors not refer to the article on The Deformable–Large Kernel Attention (DLKA) https://doi.org/10.48550/arXiv.2309.00121 and The Large Separable Kernel Attention (LSKA) https://doi.org/10.48550/arXiv.2309.01439   2. At the moment, it seems that the authors took a ready-made neural network and existing models and applied them to a specific task. It is not clear what does it add to the subject area compared with other published material. The originality of the presented models cannot be assessed. To be able to do this, authors should more clearly indicate their personal contributions in all sections of the article. Did the authors simply take ready-made models and apply them to a specific task? What new methods and models have been developed? What was integrated, and how was it done? Is the authors' results open-source, and is there a link to the repository? Is this research related to the project https://github.com/z1069614715/objectdetection_script/blob/master/yolo-improve/yolov8-project.md ?   3. The authors provided many references in the “related work” section ([2]-[15]). The authors should perform a comparative analysis of the proposed models with the mentioned references, including a quantitative analysis in the “Experiment” section.

Round 2

Reviewer 2 Report

Comments and Suggestions for Authors

In general, the authors responded to the comments indicated in the first round, but minor changes to the description of the methodology still need to be made.

It is better to cite references 1 and 2 not in the abstract but in section 3 of the article since the abstract is not a very appropriate place for references. It is also not clear why the authors use the reference to LSKA not after mentioning LSKA, but after mentioning SWDA. In the reviewer's opinion, references to ALL (not just those mentioned by the reviewer) methods and sources used should be indicated in each of the subsections of the "Method" section.
